# Within-host evolution of *Helicobacter pylori* shaped by niche-specific adaptation, intragastric migrations and selective sweeps

Florent Ailloud[1,2,3,4], Xavier Didelot[5,6], Sabrina Woltemate[2], Gudrun Pfaffinger[1], Jörg Overmann [4,7], Ruth Christiane Bader[1,8], Christian Schulz[9,10], Peter Malfertheiner[9,10] & Sebastian Suerbaum[1,2,3,4,8]

The human pathogen *Helicobacter pylori* displays extensive genetic diversity. While *H. pylori* is known to evolve during infection, population dynamics inside the gastric environment have not been extensively investigated. Here we obtained gastric biopsies from multiple stomach regions of 16 *H. pylori*-infected adults, and analyze the genomes of 10 *H. pylori* isolates from each biopsy. Phylogenetic analyses suggest location-specific evolution and bacterial migration between gastric regions. Migration is significantly more frequent between the corpus and the fundus than with the antrum, suggesting that physiological differences between antral and oxyntic mucosa contribute to spatial partitioning of *H. pylori* populations. Associations between *H. pylori* gene polymorphisms and stomach niches suggest that chemotaxis, regulatory functions and outer membrane proteins contribute to specific adaptation to the antral and oxyntic mucosa. Moreover, we show that antibiotics can induce severe population bottlenecks and likely play a role in shaping the population structure of *H. pylori*.

[1] Department of Medical Microbiology and Hospital Epidemiology, Max von Pettenkofer Institute, Faculty of Medicine, LMU Munich, 80336 Munich, Germany. [2] Institute of Medical Microbiology and Hospital Epidemiology, MHH Hannover Medical School, 30625 Hannover, Germany. [3] DZIF German Center for Infection Research, Munich Site, Munich, Germany. [4] DZIF German Center for Infection Research, Hannover-Braunschweig Site, Hannover, Germany. [5] School of Life Sciences, University of Warwick, Coventry CV4 7AL, UK. [6] Department of Statistics, University of Warwick, Coventry CV4 7AL, UK. [7] Leibniz Institute DSMZ - German Collection of Microorganisms and Cell Cultures, 38124 Braunschweig, Germany. [8] National Reference Center for Helicobacter pylori, Max von Pettenkofer Institute, 80336 Munich, Germany. [9] Department of Gastroenterology, Hepatology and Infectious Diseases, Otto von Guericke University, 39106 Magdeburg, Germany. [10] Department of Medicine 2, University Hospital, LMU Munich, 81377 Munich, Germany. Correspondence and requests for materials should be addressed to S.S. (email: suerbaum@mvp.uni-muenchen.de)

*H*elicobacter pylori is a Gram-negative bacterial pathogen that infects more than half of the human population worldwide. It is usually acquired during childhood and able to establish lifelong chronic infection. Infected patients are asymptomatic in most cases but clinical complications can range from gastric or duodenal ulcers to gastric atrophy and, ultimately, gastric cancer or mucosa-associated lymphoid tissue (MALT) lymphoma[1]. The likelihood of clinical sequelae has been linked to specific virulence factors of *H. pylori*, most importantly the *cag* pathogenicity island (*cag*PAI) and the VacA cytotoxin[2–4]. At present, recommended treatment strategies usually comprise two antibiotics, a proton-pump inhibitor, and, in some regimens, bismuth salts[5]. These treatment protocols achieve eradication rates ~ 90%[6]. Nonetheless, increasing antibiotic resistance has been reported in *H. pylori*[7], similar to other multi-drug-resistant bacteria[8].

*H. pylori* is characterized by an exceptionally high genetic diversity and variability[9,10]. The underlying mechanisms include relatively inefficient DNA repair mechanisms as well as natural competence for transformation and the ability to integrate small fragments of homologous DNA into the chromosome[9,11,12]. The species is divided into large phylogeographic populations with distinct geographical distributions, allowing the use of *H. pylori* genotypes to retrace historical human migration events[13,14].

The rates and patterns of genetic change of the *H. pylori* genome during chronic infection have been the subject of several studies, based on increasing amounts of sequencing data[11,15]. In recent years, whole-genome sequence comparisons have been used extensively to study the in vivo evolution of *H. pylori* during the early stages of infection in challenged human volunteers[16–18], in the course of long-term colonization in chronically infected individuals[16] and after intra-familial transmission[19–22]. Genes encoding outer membrane proteins (OMPs) appear to evolve faster than genes of other functional categories, suggesting adaptation to individual hosts[16]. Overall, these studies consistently determined a mutation rate of the order of $10^{-5}$ mutations per site per year.

The extent of the genetic diversity of the *H. pylori* population within a single infected stomach at a given time is much less characterized, at least compared with other bacterial pathogens[10]. The only known natural reservoir of *H. pylori* is the human stomach (including areas of gastric metaplasia in the duodenum). It is divided into three main anatomical regions: the pyloric antrum, connected to the duodenum; the corpus in the center and the fundus in the upper curvature, next to the cardia. The analysis of single isolates from separate gastric regions revealed that strains from the same host can be genetically differentiated[16,19,23–25]. Stomach regions contain different human cell types providing distinct environmental conditions for *H. pylori* subpopulations, and these could lead to specific niche adaptation. For example, the differences in pH between antrum and corpus were shown to select for variant BabA adhesins with different kinetics of pH-dependent binding affinities[26]. Differential antibiotic resistance profiles have also been observed in related strains isolated in the antrum and in the corpus[27], suggesting that *H. pylori* populations can be structured by the stomach anatomy.

The time since the most recent common ancestor (TMRCA) between isolates from the same stomach has been calculated from genomic data using pairs of strains isolated from antrum and corpus. Surprisingly, TMRCA values appeared to be independent of the host's age and were only a few years on average[19]. Mutation rates calculated in acute and chronic phases of the infection are comparable[15,16,18], thus the TMRCA of a *H. pylori* population was expected to increase linearly beginning at the infection during childhood. Population bottlenecks and strain turnovers are possible explanations for the low average TMRCA, which could be induced by selective pressures from the host immune system as well as common antibiotic consumption.

The aim of this study was to characterize *H. pylori* population diversity and dynamics within the stomach using bacterial genomics. Multiple single colonies of *H. pylori* were isolated from several gastric regions in naturally infected adults and analyzed by whole-genome sequencing. Our analyses reconstruct the pathways of migration of bacterial subpopulations between parts of the stomach, and identify several candidate genes with high frequencies of within-host variation and/or genetic signals of association to specific regions of the stomach, suggesting these loci participate in local adaptation to gastric niches.

## Results

**Extensive within-host diversity in *H. pylori* populations.** In order to quantify the genome-wide diversity of *H. pylori* within the human stomach, gastric biopsies were obtained from 16 *H. pylori*-infected individuals (mean age, 49.9 years, nine women, seven men, Table 1). Twelve of the 16 patients displayed chronic gastritis in histology, whereas four had atrophic gastritis, including one patient with intestinal metaplasia. Five patients had a known history of one or multiple attempted eradication therapies. Gastric biopsies were taken from the antrum, corpus, and fundus regions for 10 of these patients. For the six remaining patients, only antrum and corpus were sampled. *H. pylori* was cultured from the biopsies, and from each biopsy, 10–15 single colonies were picked, propagated, and the genome sequences of these clones determined with an Illumina MiSeq instrument. As this short-read sequencing approach resulted in draft sequences consisting of 39 contigs on average, we selected 1–3 isolates from every patient, and additionally sequenced their genomes with Single Molecule Real-Time (SMRT) technology, which yields longer reads and permits to assemble a closed complete-genome sequence. The total data set consisted of genomes from 440 *H. pylori* isolates, including 414 draft sequences plus 26 closed sequences (Supplementary Data 1).

All isolates from the 16 patients could be assigned to the phylogeographic population hpEurope. The average within-host pairwise nucleotide identity ranged from 98.5 to 99.8%, and isolates from a given patient clustered tightly together in a global phylogenetic tree (Fig. 1a), suggesting we did not sample any mixed infection with unrelated *H. pylori* strains. The amount of genetic diversity observed within the group of *H. pylori* isolates from one individual patient was highly variable between patients (Fig. 1b). More than 18,000 unique SNPs were identified within the most diverse population (clones isolated from patient 476), whereas only 73 SNPs were found in the least diverse population (from patient 173).

Out of 16 patients, *cag*PAI-positive and *babA*-positive strains were isolated from 10 and 12 patients, respectively. All *cag*PAI-positive patients were *babA*-positive as well. All BabA sequences had residues in position 198 associated with generalist binding preference[28], except in patient 23 that had a glutamine typically found in specialist strains. In patient 476, related *cag*PAI-positive and -negative isolates were identified. Both *cag*PAI+ and *cag*PAI− isolates were distributed in several clades with no association with gastric regions. CagA was still present in all isolates, and a second copy with two additional EPIYA-C motifs was found in one of the several *cag*PAI-positive clades of this patient (Table 1). The microdiversification of the *cag*PAI in this patient suggests a complex history of gene gain and loss driven by fluctuating selective pressures within different niches of the stomach and likely made possible by secondary infections.

**Table 1 Overview of *H. pylori* populations and their major characteristics**

| Patient | Age (y) | TMRCA (y) | Patho.[a] | Isolates (A-C-F) | Treatment[b] (ATB/PPI) | CLR[c] | CIP[c] | MTZ[c] | Unique SNPs | r/m[d] | Rec. events[e] [length(kbp)] | cagPAI |
|---|---|---|---|---|---|---|---|---|---|---|---|---|
| 173 | 44 | 0.34 | CG | 10-10-NA | Y/N | R | R6S14 | R | 73 | 1.2 | 11[0.2] | + |
| 478 | 70 | 1.59 | AG/IM | 10-10-NA | Y/Y | R | R | R | 225 | 0.3 | 7[0.7] | + |
| 280 | 76 | 0.51 | CG | 10-10-NA | Y/Y | S | R | R | 87 | 0.9 | 7[0.4] | − |
| 169 | 50 | 2.09 | CG | 10-10-NA | N/N | R | S | S | 1919 | 9.0 | 109*[45] | − |
| 476 | 34 | 7.19 | CG | 15-15-NA | N/N | S | S | S | 18147 | 16.3 | 1126*[423.6] | +/− |
| 479 | 69 | 4.5 | AG | 15-15-NA | N/N | S | S | S | 13295 | 15.7 | 562*[321.3] | + |
| 21 | 58 | 4.04 | CG | 10-10-10 | N/N | S | S | S | 1066 | 0.6 | 53[3.1] | − |
| 381 | 42 | 1.5 | AG | 10-10-10 | Y/N | R21S9 | S | R10S20 | 503 | 1.0 | 20[2.6] | − |
| 5 | 49 | 3.24 | CG | 10-10-10 | N/N | S | S | R | 849 | 0.6 | 39[2.6] | − |
| 26 | 47 | 1.57 | AG | 10-10-10 | NA/N | S | S | S | 409 | 0.4 | 4[0.4] | + |
| 23 | 30 | 1.87 | CG | 10-10-10 | N/N | R23S7 | S | R20S10 | 424 | 0.3 | 15[0.6] | + |
| 13 | 54 | 3.45 | CG | 10-10-10 | N/N | S | S | S | 1108 | 0.8 | 45[4.8] | + |
| 20 | 55 | 1.27 | CG | 10-10-10 | Y/Y | R | S | R14S16 | 415 | 0.9 | 22[1.7] | + |
| 24 | 33 | 3.29 | CG | 10-10-10 | N/N | S | S | S | 1156 | 0.3 | 29[1.2] | + |
| 25 | 56 | 3.31 | CG | 10-10-10 | NA/N | S | S | R18S12 | 3684 | 9.2 | 135*[87.1] | − |
| 19 | 32 | 3.21 | CG | 10-10-10 | N/N | S | S | S | 2632 | 3.3 | 177*[33.4] | + |

[a]CG: chronic gastritis; AG: atrophic gastritis; IM: intestinal metaplasia
[b]Treatments undergone by the patient prior to the endoscopy. ATB: antibiotic-based eradication. PPI: proton-pump inhibitor
[c]CLR: Clarithromycin; CIPRO: Ciprofloxacin; MET: Metronidazole. Distribution of resistant (R) and susceptible (S) genotypes are indicated for heteroresistant populations
[d]r/m represent the ratio of SNPs introduced by recombination versus mutation
[e]Total number of recombination events with the cumulative length (kbp) of recombined segments shown in brackets. Stars indicate that the majority of recombination events in this population are likely to have been imported during a period of mixed infection

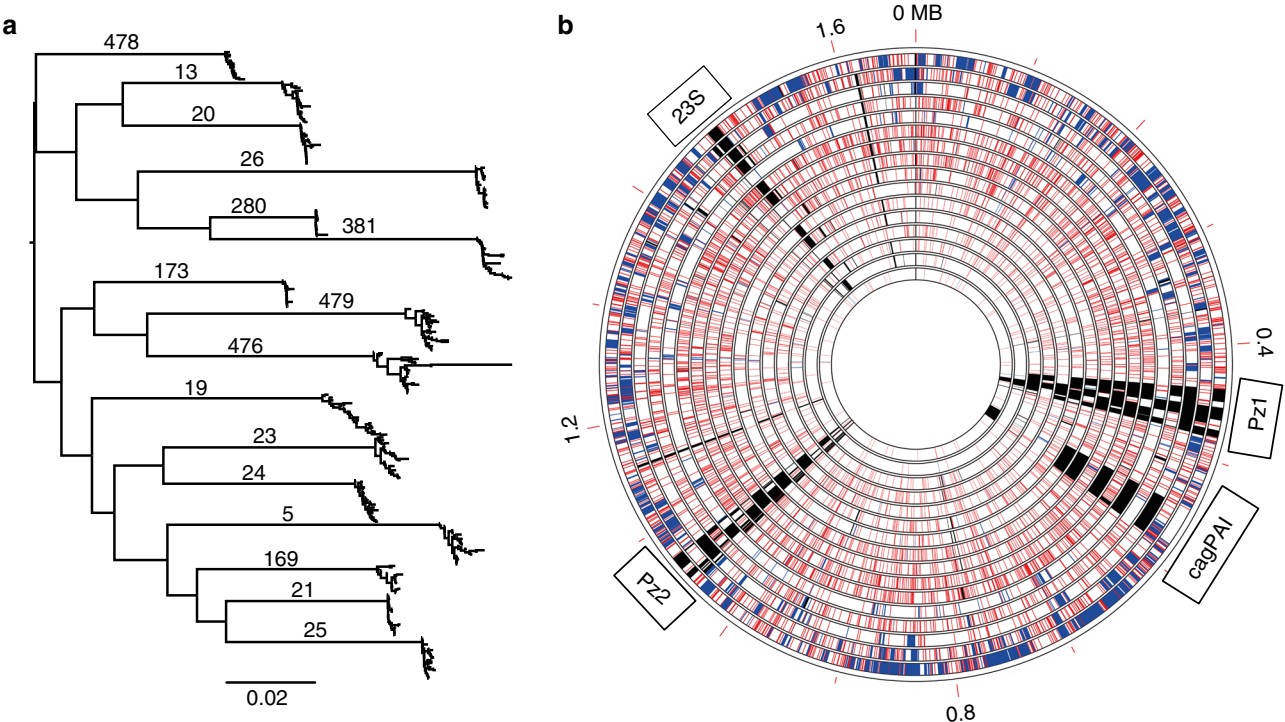

**Fig. 1** Genetic diversity within *H. pylori* populations from 16 human individuals. **a** Global phylogeny including 440 *H. pylori* isolates from 16 human individuals. Isolates from the same patient are clustering together tightly indicating that no mixed infection was sampled. The scale bar represents the number of substitutions per site. Source data are provided as a Source Data file. **b** Every circle represents the *H. pylori* population from one individual patient. Each population consists of 20–30 clones (see text for details). Circles are ordered by decreasing genetic within-host diversity from outer to inner circle. From the outer to inner ring: patient 476, 479, 25, 19, 169, 24, 21, 13, 5, 381, 23, 26, 20, 478, 173, 280. Genetic coordinates were synchronized across all populations by mapping onto the reference strain 26695. Spontaneous mutation and recombination events are represented by red and blue lines, respectively. Regions present in 26695 but not fully covered in a population are indicated by black lines (the 23S region indicated here corresponds to the second copy, which is not always properly assembled in draft genomes, resulting in uncovered regions)

Variations of gene content and sequences are not the only ways *H. pylori* can evolve within its host. We identified large genomic rearrangements between the three closed genomes of patient 381 generated by SMRT sequencing (Supplementary Fig. 1a). Recombination between chemotaxis genes *tlpA* and *tlpC* induced the inversion of a small 9 kb fragment, whereas recombination between the 23S rRNA gene copies resulted in the inversion of a much larger 953 kb region. Finally, two inversions of 81 kb and 643 kb were owing to an exchange between copies of the miniIS605, a non-autonomous transposable element found in most *H. pylori* isolates that contains the non-coding RNA HPnc0580[29,30]. In the two reference genomes of patient 25, an inversion of ~ 590 kb owing to recombination between HP0488-HP1116 paralogs was also observed (Supplementary Fig. 1b). No rearrangements were observed in the other five patients where multiple closed genomes were available.

**Bacterial migration between gastric regions**. Most previous studies about *H. pylori* within-host diversity are restricted to the antrum and corpus regions. Here, fundus biopsies were also obtained for 10 of the 16 patients. In order to observe how the genetic diversity of each *H. pylori* population is distributed across the gastric regions, we computed recombination-corrected phylogenetic trees from whole-genome alignments. A visual inspection of these trees revealed that a structure with multiple clusters was clearly apparent in most patients. The degree to which these clusters correlated with the location from which the bacteria had been sampled was variable, and isolates with different origins were frequently found within the same cluster (Fig. 2).

This observation can be interpreted in terms of migration of *H. pylori* between the antrum, corpus, and fundus regions. We used ancestral state reconstruction methods to reconstruct the intra-stomach migration routes of individual *H. pylori* isolates, as they last shared a common ancestor within a patient. By comparing three migration models of increasing complexity (see Methods) we found evidence for differences of the migration rates between parts of the stomach. There was no evidence for asymmetry in the

rates between any two parts. The data fit well to the symmetric model, in which three parameters are estimated between antrum and fundus, between antrum and corpus, and finally between corpus and fundus. Migration events between corpus and fundus were significantly (based on 95% confidence intervals of the maximum likelihood estimates) more frequent than between antrum and corpus (Fig. 3a). The migration rate between antrum and fundus was negligible. Following these observations, we tested one additional meristic model where migration from antrum to fundus cannot happen directly but only in two steps via the corpus, and this model had a slightly better goodness of fit (AICc = 546.51) than the symmetric model (AICc = 548.55). Finally, we tested an alternative approach, stochastic character mapping (see Methods), to simulate the migration along the population trees and obtain the number of transitions between each region. Following the same trend as the ancestral reconstruction method, this approach showed that the majority of the migration events happened between corpus and fundus, whereas migration between antrum and fundus represent the minority (Fig. 3b). Overall, these results suggest that *H. pylori* gastric subpopulations are typically segregated in the antrum mucosa but are migrating more freely between regions lined by oxyntic mucosa, which includes both corpus and fundus.

We next focused on variation between patients, and to do so, repeated the ancestral state reconstruction analysis separately for each patient's *H. pylori* population. In five patients (19, 20, 21, 23, 381), the meristic model was the best fit, with more frequent exchange between corpus and fundus than between antrum and corpus, similar to results obtained on the overall data set. The isolates of patient 26 also fit the meristic model but displayed more exchanges between antrum and corpus. However, this population was hardly differentiated according to the topology of its phylogenetic tree with no clearly identifiable clusters, suggesting that this result may be artefactual. In the four remaining patients (5, 13, 24, 25) the equal rates (ER) model had the best fit, suggesting that migration is less influenced by gastric regions for these strains. However, these results are likely explained by the low total number of migration events in most of

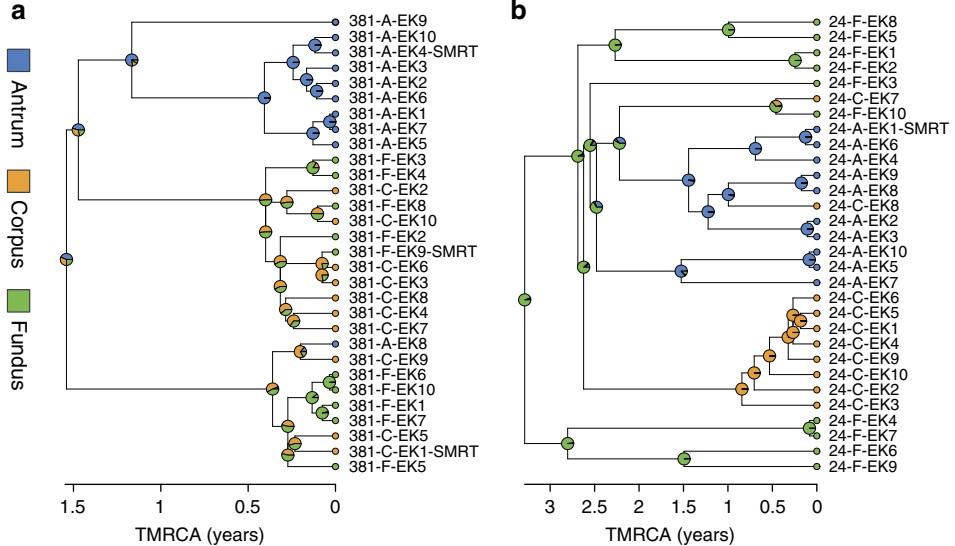

**Fig. 2** Clustering structure of the *H. pylori* population varies between individual patients. Representative phylogenetic trees of within-host *H. pylori* populations displaying different degrees of clustering. Trees were generated as described in the methods (source data are provided as a Source Data file). The time since the most recent common ancestor (TMRCA), expressed in years, is represented by the X axis. Pie charts correspond to the posterior probability distribution of ancestral characters (i.e., likelihood that ancestral lineages originated from antrum: blue, corpus: orange or fundus: green). **a** Phylogenetic tree of patient 381 isolates showing a well-defined cluster structure. **b** Phylogenetic tree of patient 24 isolates showing a poorly defined cluster structure

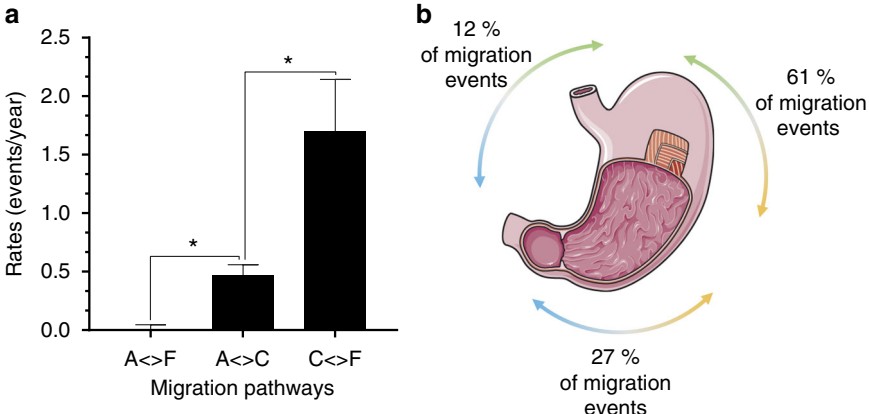

**Fig. 3** *H. pylori* preferentially migrates between the corpus and fundus regions of the stomach. **a** Estimation of symmetric migration rates by marginal state reconstruction based on a global structure constructed by concatenating individual time-scaled trees (A: antrum, C: corpus, F: fundus; *: *p* < 0.05). Error bars represent 95% confidence intervals of the maximum likelihood estimates. **b** Schematic representation of the migration events distribution predicted by stochastic character mapping in relation to the stomach anatomy (blue: antrum, orange: corpus, green: fundus). Source data are provided as a Source Data file. Panel 3b is a derivative of "Stomach 4" by Servier Medical Art, used under CC BY 3.0 [https://creativecommons.org/licenses/by/3.0/]

these patients, so that there is a lack of statistical power to determine the inequality of the rates of migrations between regions. Overall, frequencies of migration did not correlate with genetic diversity, indicating that the variation we observed between populations is not explained by the difference between their time of divergence and that this phenomenon is also influenced by host genetics and environment.

Despite the extensive mixing between fundus and corpus, the fundus isolates added a significant amount of diversity to our data set, therefore subsequent analyses in this study were only carried out on the subset of 10 patients including fundus isolates in order to reduce bias and to obtain a homogeneous sampling.

**Frequent within-host variation in specific functional groups**. Within-host diversification is the net result of both random genetic drift and recurrent selection. The low levels of variability in some populations combined with the high levels of recombination in others make it challenging to quantify the contribution of each evolutionary mechanism reliably. We cross-compared genes that contained non-synonymous SNPs (NS-SNPs) between isolates from each patient, with the aim to determine whether specific loci were frequently and independently targeted by positive selection. We observed a high frequency of within-host variation in 40 genes (referred to later as host-variable genes) containing NS-SNPs in more than half of the patients, which was significantly higher (based on 95% confidence intervals of simulated random sampling, 10,000 replications) than expected if polymorphisms were randomly distributed in the genome (Table 2, Supplementary Fig. 2). OMPs were over-represented, making up 28% of the 40 host-variable genes. In particular, two OMPs from the Hop family, HopB (AlpA) and HopL, were variable in all the patients. The cellular processes as well as the transport and binding protein functional categories were also over-represented at 18% and 15%, respectively (Supplementary Data 2). Genes coding for the VacA cytotoxin as well as two of the toxin-like paralogs VlpC and ImaA were variable in 70–90% of the patients. Although most of the host-variable genes displayed only two or three allelic forms per patient, four to five variants were observed for *hopL*, *hopZ*, *tlpB*, *rpoB,* and HP1322. Finally, recurrent identical mutations were frequently observed, with 70% of the 40 host-variable loci containing polymorphisms found in multiple patients, which could be evidence of parallel adaptation.

**Niche-specific evolution and adaptation**. *H. pylori* is able to colonize the entire stomach despite the differing environmental conditions existing between gastric niches. The selective pressures caused by these differences might explain why specific genes evolve frequently within one host. The restricted migration of antrum isolates could also be a consequence of region-specific adaptation. Similarly, the frequent exchanges between corpus and fundus could reflect the similarities between these two regions. Consequently, we sought to determine whether polymorphisms were associated with the antrum, compared with the oxyntic mucosa (i.e., corpus and fundus), or vice-versa, using a Fisher's exact test. Furthermore, we hypothesized that recently migrated isolates are likely to retain the genetic background representative of their original gastric region temporarily, which could introduce noise in the association test. To account for this, we used our ancestral state reconstruction analysis to infer recent migration events. When possible, we re-assigned the location of each isolate using the location of the corresponding ancestral node with a maximum divergence time of 6 months.

We identified 1071 polymorphisms that had significant associations with gastric regions (false discovery rate (FDR) adjusted *p* value < 0.01). After discarding synonymous and recombined polymorphisms, 309 non-synonymous mutations remained within a total of 232 genes (referred to later as niche-specific mutations and genotypes, respectively). Although we observed 32 niche-specific genotypes per patient on average, no significant results were obtained in patients 25 and 26. Isolates from these two patients appear shuffled together with barely any region-specific clusters, suggesting there might be limited adaptation and free migration in these patients. In addition, only one genotype with significant niche association was found in patient 24 despite clearly apparent region-specific clusters. No recent migration events were corrected by ancestral reconstruction because of the deep branches of the star-like phylogeny in this patient. In particular, one isolate sampled in the corpus was located on a long branch within a cluster containing 70% of the antrum isolates indicating a likely origin from the corpus. By increasing the maximum divergence time for re-assignment from 6 months to 1 year for patient 24, we identified nine additional genotypes with association to gastric regions (Supplementary Data 3). The gene that contained the most mutations associated with gastric regions encodes the fused beta and beta' subunits of the DNA-dependent RNA polymerase RpoB. Niche-specific mutations were found in *rpoB* in five patients (5, 13, 19, 20,

**Table 2 High-frequency host-variable genes within 10 *H. pylori* populations**

| Counts | LocusTag | Gene | Product |
|---|---|---|---|
| 10 | HP0913 | *hopB* | outer membrane protein (AlpA) |
| 10 | HP1157 | *hopL* | outer membrane protein |
| 9 | HP0099 | *tlpA* | methyl-accepting chemotaxis protein |
| 9 | HP0289 | *ImaA* | immunomodulatory autotransporter, vacA-like protein |
| 9 | HP0953 | - | hypothetical protein |
| 8 | HP0009 | *hopZ* | outer membrane protein |
| 8 | HP0782 | *hofE* | outer membrane protein |
| 8 | HP1156 | *hopI* | outer membrane protein |
| 8 | HP0103 | *tlpB* | methyl-accepting chemotaxis protein |
| 8 | HP0599 | *tlpD* | methyl-accepting chemotaxis protein |
| 8 | HP0922 | *vlpC* | toxin-like outer membrane protein |
| 8 | HP0130 | - | putative uncharacterized protein |
| 8 | HP1198 | *rpoB* | DNA-directed RNA polymerase subunit beta–beta' |
| 8 | HP0160 | *hcpD* | cysteine-rich protein D |
| 8 | HP1117 | *hcpX* | cysteine-rich protein X |
| 7 | HP0021 | *lpxE* | lipid A 1-phosphatase |
| 7 | HP0252 | *hopF* | outer membrane protein |
| 7 | HP0655 | *bamA* | outer membrane protein assembly factor |
| 7 | HP0912 | *hopC* | outer membrane protein (AlpB) |
| 7 | HP0887 | *vacA* | vacuolating cytotoxin |
| 7 | HP1322 | - | putative uncharacterized protein |
| 7 | HP0164 | *arsS* | sensor histidine kinase |
| 7 | HP0235 | *hcpE* | cysteine-rich protein E |
| 6 | HP0407 | *bisC* | biotin sulfoxide reductase |
| 6 | HP0772 | *amiA* | N-acetylmuramoyl-L-alanine amidase |
| 6 | HP0805 | *lex2b* | LPS biosynthesis protein |
| 6 | HP0254 | *hopG* | outer membrane protein (omp8) |
| 6 | HP1453 | *homD* | outer membrane protein |
| 6 | HP0392 | *cheAY2* | chemotaxis sensor kinase |
| 6 | HP0499 | *pldA* | phospholipase A1 |
| 6 | HP0219 | - | hypothetical protein |
| 6 | HP0586 | - | hypothetical protein |
| 6 | HP1039 | - | O-antigen polymerase |
| 6 | HP1021 | - | orphan response regulator |
| 6 | HP0818 | *proWX* | osmoprotection protein |
| 6 | HP1251 | *oppB* | ABC transporter, permease protein |
| 6 | HP1252 | *oppA* | ABC transporter, oligopeptide-binding protein |
| 6 | HP0471 | *kefB* | potassium-efflux protein |
| 6 | HP0876 | *frpB1* | iron-regulated outer membrane protein |
| 6 | HP1400 | *fecA3* | iron(III) dicitrate transport protein |

Genes containing within-host NS-SNPs were cross-compared in the subset of 10 patients including fundus isolates. Genes were considered host-variable when containing NS-SNPs in more than half of the tested populations (counts are indicated in the first column). Locus tags from 26695 are given as reference

21), but none were located at the same position. They were specifically centered around the beta–beta' interface region and differed from mutations described previously to be involved in rifabutin resistance or Western/East-Asian geographical association[31,32]. Globally, adaptation to gastric regions appeared to follow a specific path in each patient, as only 17% of the niche-specific genotypes were found recurrently in multiple patients. Niche-specific genotypes were not necessarily overlapping with the high-frequency host-variable genes identified previously. About 20% of the host-variable OMPs did not display any sign of adaptation. Despite being polymorphic in 10 patients, *hopB* and *hopL* contained niche-specific genotypes in only two and one patients, respectively. The absence of association to gastric regions in host-variable loci could result from antigenic variation or indicate adaptation to a different gastric niche. For example, three out of the six members of the Sel1-like repeat (SLR) cysteine-rich protein family were host-variable in seven to eight patients but only had niche-specific genotypes in one to two: *hcpD* (HP0160), *hcpE* (HP0235), and *hcpX* (HP1117) (Table 2, Supplementary Data 3). All mutations were located within SLR repeats, which are numerous in these proteins. Proteins of this family are known for being highly immunogenic[33,34] thus

mutations could contribute to reducing inflammation. Despite belonging to the same family these genes are located far apart on the chromosome, supporting the hypothesis that these loci may have been under selection by common environmental conditions.

Because *H. pylori* populations are inherently structured by gastric regions, it cannot be excluded that niche-specific mutations emerged purely from genetic drift. To pinpoint niche-specific genotypes, which were more likely to be involved in adaptation, we performed functional enrichment analyses and tested for natural selection using a McDonald–Kreitman test[35]. Genes belonging to the OMPs, chemotaxis and motility, toxin production, and regulatory functions functional categories, were significantly over-represented (Chi-square, $p < 0.05$) (Fig. 4). In total, 66% of these genes showed a significant (Fisher's exact test, FDR adjusted $p$ value $< 0.05$) departure from neutrality, with a neutrality index (NI) $> 1$ (when it could be calculated). Here, NI $> 1$ indicates an excess of within-host non-synonymous poly-morphisms. Considering that niche-specific mutations are fixed within local subpopulations, an excess of within-host non-synonymous substitutions can be interpreted as a sign of local adaptation. Unlike what we observed in high-frequency host-variable genes, niche-specific mutations were always unique to a

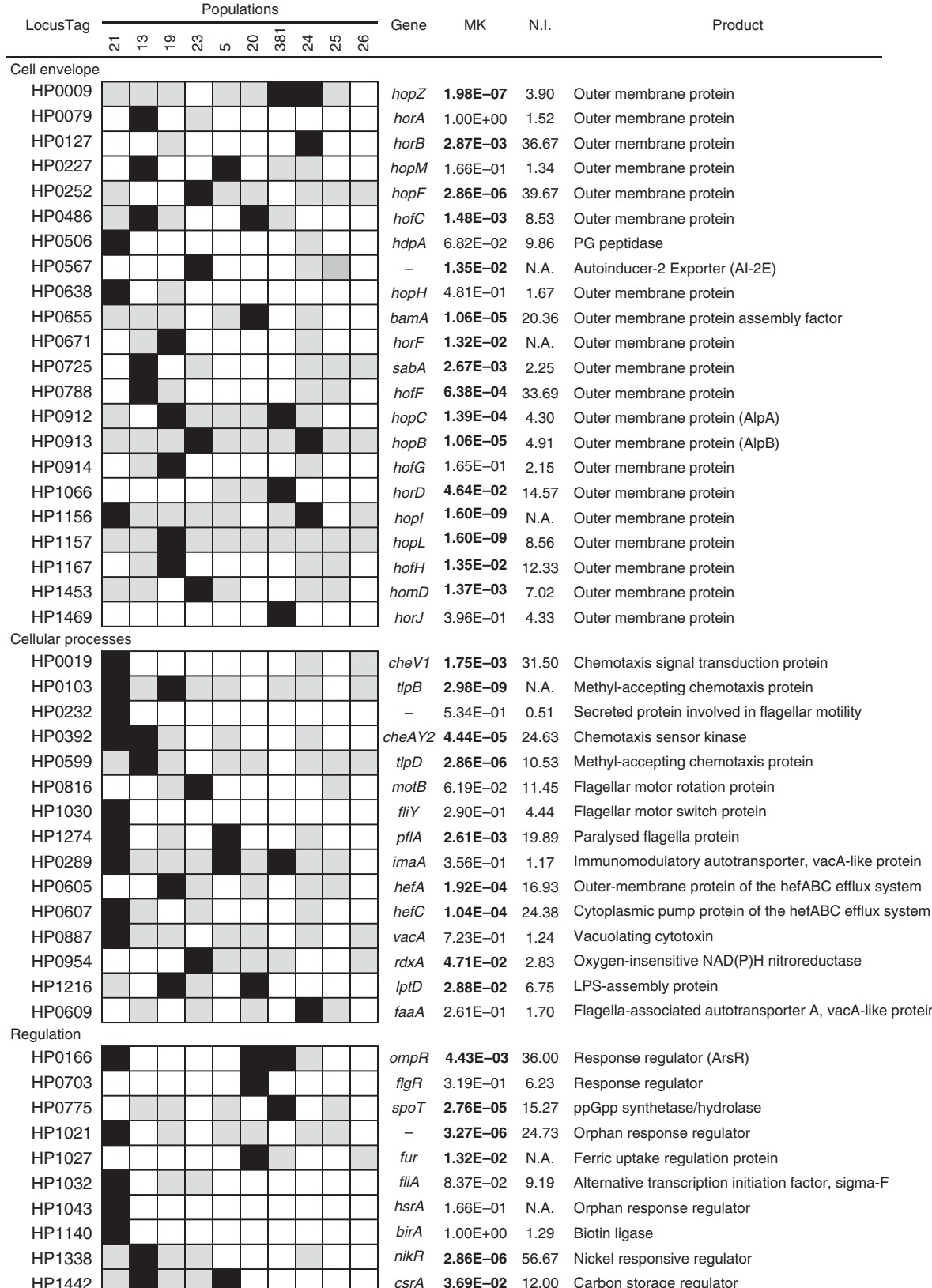

**Fig. 4** Evidence of natural selection in niche-specific genotypes from enriched functional categories. For each *H. pylori* population, black and gray squares represent polymorphic genes with and without association to gastric regions, respectively; white squares indicate genes without non-synonymous mutations. Locus tags from *H. pylori* strain 26695 are given as reference. MK indicates the Fisher's exact FDR adjusted *p* value from a McDonald–Kreitman test (bolded values indicate significance at *p* < 0.05). N.I.: Neutrality Index

patient, except in the *hofC* OMP. HofC displayed a significant departure from neutrality and an isoleucine to valine mutation in position 206 was observed in four patients (13, 19, 20, 21). In two of these (13 and 20), this mutation was found associated to gastric regions with the valine residue found predominantly in the antrum isolates of both patients.

Several genes related to the regulation of motility through chemotaxis contained niche-specific mutations with evidence of natural selection. The *cheAY2* gene (HP0392) codes for a bifunctional protein containing both parts of two-component regulatory system: CheA (histidine kinase sensor) and CheY (response regulator)[36]. All niche-specific residues were located in the CheA region of the protein. Three out of the four methyl-accepting chemotaxis proteins: TlpA (HP0099), TlpB (HP0103), TlpD (HP0399) were host-variable in eight to nine patients, but only TlpB and TlpD had niche-specific mutations in two and one patients, respectively. Around 70% of the mutations were clustered close to the doubleCache signaling domain in TlpA but none were associated with gastric regions. Mutations were more homogeneously distributed along the sequence in TlpB and TlpD, but the niche-specific mutations were located in or close to the methyl-accepting domain.

**Mosaic sequences owing to inter-strain recombination.** *H. pylori* can quickly exchange substantial parts of its genome when exposed to DNA from another *H. pylori* strain owing to its natural competence[11,16]. Nonetheless, it colonizes a very isolated niche and no natural reservoir outside of the human stomach is known, thus the only opportunity for inter-population recombination is during mixed infections. Such sudden increase of genetic diversity might reshape the evolutionary dynamics of a population; thus we applied ClonalFrameML[37] to analyze the effect of recombination in each patient. Recombined fragments were observed in the *H. pylori* populations from all patients, but large differences were observed in the relative contributions of recombination and mutation to the overall diversity, as well as the number of events and average and total recombined length. For 11 out of 16 patients, manual curation revealed that a majority of the recombination events were located in duplicated genes or paralogous families. Sequences identical to the recombined segments could be frequently found in the other copies of the gene within the same isolate, suggesting these are the products of intragenomic recombination. The most frequently involved genes were *babABC* and *sabAB*, the *cag* pathogenicity island gene *cagY* and the OMP gene *hopQ*.

Only related isolates were found at the time of the endoscopies, suggesting transient mixed infections can quickly increase the genetic diversity of the main population. The r/m ratio of the five patients (19, 25, 169, 476, 479) with clear evidence of past mixed infections ranged from 3 to 16. Imports were $465 \pm 185$ bp long on average and replaced 2–26% of the genome at the population level. Overall, an average 80% of the predicted imports introduced at least one NS-SNP and thus had the potential to modify protein functions. Further manual curation revealed that many imports overlapped the same region in multiple isolates from the same patient but had different sizes. We observed three patterns of overlapping imports (Supplementary Fig. 3). The first pattern suggests that some imports may not originate directly from a mixed infection (primary import) but rather from previously transformed isolates within the same population (secondary import). This model is further supported by the phylogenetic distribution of these imports, because primary and secondary imports were found in polyphyletic clusters indicating they likely originated from multiple events. Although the dating of internal nodes is not fully accurate owing to the limited number of

isolates, nodes containing primary imports typically preceded nodes containing the secondary imports. The second pattern we observed suggest that primary imports can be partially reversed by recombination with a wild-type isolate from the same population (reversed import). Reversed imports could be easily identified when the primary imports are still intact in other isolates with a direct common ancestor. The third pattern was observed less frequently and corresponds to multiple independent primary imports targeting the same region. These imports are overlapping but do not share any border. They are located in polyphyletic clusters and have slightly different polymorphisms, suggesting they originated from distinct lineages of the mixed infection.

**Effect of antibiotics on *H. pylori* population diversity.** Although *H. pylori* infection usually begins during childhood, previous studies performed with paired isolates from antrum and corpus showed that the within-host TMRCA of *H. pylori* is typically low (~1–5 years). To determine the TMRCA values in our data set, we used ClonalFrameML[37] to generate phylogenetic trees adjusted for recombination. Trees were then scaled to time using a fixed mutation rate of $1.38 \times 10^{-5}$ per site per year[19]. The average TMRCA of the *H. pylori* populations from the 16 patients was 2.5 years (0.34–7.17). No correlation with patient age was observed. For example, a TMRCA of 4 months was calculated for a 44-year old woman (patient 173), whereas a TMRCA of 7 years was estimated for a 34-year old man (patient 476). Several hypotheses have been proposed to explain low TMRCA, including strong genetic drift or diversity bottlenecks induced by immunity or antibiotics. As five patients reported at least one attempted eradication therapy prior to the gastroscopy, we explored the role of antibiotics in shaping the population structure of *H. pylori* within a stomach.

Clarithromycin (CLR), ciprofloxacin (CIP), and metronidazole (MTZ) are antibiotics used for the eradication of *H. pylori* as well as for treatment of other pathogenic bacteria. The emergence of resistant phenotypes has been associated with specific genotypes: two specific positions of the 23S rRNA gene are responsible for high or intermediate levels of resistance to CLR; mutations in the gene encoding the DNA gyrase subunit A (*gyrA*) can confer resistance to CIP, whereas resistance to MTZ have been associated with either mutations or gene inactivation of the nitroreductase gene *rdxA*[38]. We searched for resistance-associated polymorphisms in the complete data set to characterize the spread of resistance within the populations in response to either *H. pylori* eradication therapy or unrelated antibiotic intake targeted toward other infections. The resistant genotypes we observed were systematically confirmed phenotypically in a subset of isolates. There was no phenotypical evidence for resistance against either beta-lactam antibiotics or rifabutin. This analysis revealed that the *H. pylori* populations from the five patients who had previously been prescribed eradication therapy (patient 20, 173, 280, 381, 478), as well as two additional patients who were not known to have undergone eradication treatment (23, 169), contained isolates that were resistant to at least one of the tested antibiotics (Table 1). The mean TMRCA of resistant populations was $1.6 \pm 1$ years, whereas the mean TMRCA for fully susceptible populations was $3.9 \pm 1.7$ years ($p < 0.05$, Welch's *t* test).

Resistance-associated alleles were often not fixed, with 20–60% of the population still carrying susceptibility-associated alleles depending on the antibiotic class. Allele fixation in some cases was limited to specific gastric regions. For example, isolates from the corpus, but not the antrum, were resistant to CIP in patient 173. Antrum isolates represented the only MTZ susceptible

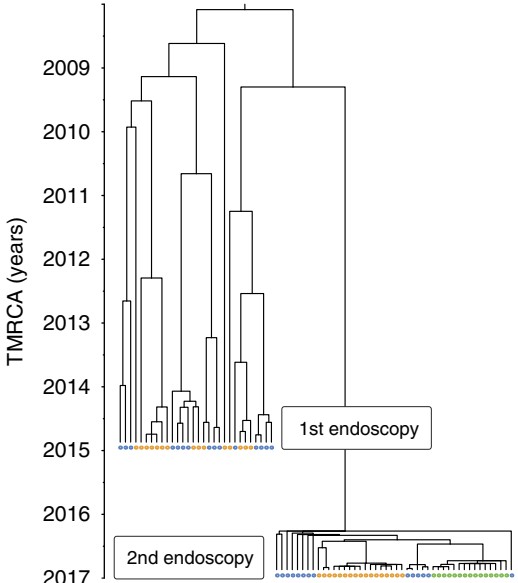

**Fig. 5** Major population bottleneck in a *H. pylori* population. Two sets of gastric biopsies were obtained two years apart in a 34 years old treatment-naive patient with chronic gastritis (patient 476). A time-scaled tree was created using the isolates obtained at both time points (blue: antrum, orange: corpus, green: fundus). The population observed at the first time point had an estimated TMRCA of 7 years. The directly related population observed at the second time point had a TMRCA < 1 year and displayed a phylogenetic structure suggesting a fast clonal expansion subsequent to a bottleneck. Source data are provided as a Source Data file

cluster in patient 23 and, on the opposite, the only resistant one in patient 20. Although the whole population in patient 280 was resistant to CIP, the isolates had a different genotype in the antrum (D91G) compared to the corpus (D91N). To further investigate the low TMRCAs of *H. pylori* populations, we obtained a new set of antrum, corpus, and fundus biopsies from patient 476, 2 years after the initial endoscopy. As before, 10 single colonies were cultured from each biopsy and sequenced on an Illumina MiSeq instrument. This patient did not decide to undergo any kind of *H. pylori* eradication regimen, leading us to believe that the *H. pylori* population evolved unhindered between the examinations. Nonetheless, the population recovered from the second set of biopsies appeared to have lost >98% of its genetic diversity (apparent bottleneck) and gone through a fast clonal expansion, all within the span of 2 years (Fig. 5). Both *cag*PAI-positive and -negative isolates were found at the first time point, however the population from the second time point was entirely *cag*PAI-negative. Moreover, we found no evidence for inter-population recombination anymore at the second time point. We calculated a TMRCA < 1 year, compared to the 7 years TMRCA of the original population. Although we did not observe resistance to CLR, CIP, or MTZ at the first time point, the population from the second time point was homogeneously resistant to CLR. This result suggests that the population was exposed to antibiotics unrelated to *H. pylori* eradication treatment between the endoscopies.

## Discussion

Although the existence of multiple *H. pylori* strains with distinguishable but related genotypes within one stomach has been reported before, the within-host diversity of *H. pylori* has not been determined in a systematic and quantitative way. Here, we used a combination of sequencing technologies to characterize the host-associated population diversity of *H. pylori* in 16

naturally infected individuals. The extent of genetic diversity observed in each *H. pylori* population from one stomach was highly variable. No ongoing mixed infection was detected, but strong evidence of recombination with one or multiple unrelated *H. pylori* strains during a past mixed infection was found in five patients. Through natural transformation, the co-existence with another population appears to have quickly increased the overall genetic diversity in these populations. Nonetheless, the isolates were not all impacted in the same way by inter-population recombination, which is in agreement with a previous study based on paired isolates from the gastric antrum and corpus regions, which showed that even in South Africa, a high prevalence country, the contribution of recombination to diversity varied widely between different individuals[19]. Individual lineages affected by recombination with a secondary infection were clearly identified and imported segments were sometimes laterally transferred to other lineages or, conversely, reverted back to wild-type by another recombination event with isolates from the original population. This phenomenon suggests that the large amount of diversity imported from a secondary infection is subsequently filtered by selection. In agreement with our previous observations[19], the TMRCAs of all populations in our data set were low and did not correlate with patients' age despite the fact that *H. pylori* infections typically are acquired during childhood[39]. Five patients had undergone one or multiple *H. pylori*-specific eradication therapies in the years preceding the gastroscopy. Independently of the outcome, antibiotic regimens have the potential to result in an apparent reduction of the genetic diversity of the population in the stomach. Successful eradication can be followed by re-infection with a new population since *H. pylori* does not induce protective immunity[40–43]. In our case, the *H. pylori* isolates in all five treated patients were resistant to one or several antibiotics commonly used for eradication therapies, suggesting that the treatment failed and only induced a population bottleneck, followed by recrudescence of the infection. Such temporary reductions of the *H. pylori* population have been termed "elimination" (in contrast to eradication) and are a likely cause of the low observed TMRCAs. However, antibiotic resistance was also observed in patients with no known history of eradication treatment, suggesting that antibiotic treatments for unrelated infections was selecting antibiotic-resistant *H. pylori*[7]. It is known from early treatment studies that monotherapies have the potential to severely reduce *H. pylori* colonization[44]. Within-host populations were not always homogeneously resistant to antibiotics, confirming previous reports of heteroresistance in *H. pylori*[45]. Although the complete fixation of resistance genotypes can be explained by intrinsic resistances or hard selective sweeps, partially fixed alleles or independent adaptation are evidence for acquired resistance and soft selective sweeps, suggesting a relatively high fitness cost of resistance. The sweeping of alleles through the population appeared to be modulated by the stomach environment, and some genotypes were restricted to specific gastric regions. In these patients, heteroresistance suggests that antibiotics exerted a weak selective pressure causing a slow spread of adaptive alleles within the stomach. Such an effect may be explained, for example, by the relative instability of clarithromycin, and macrolide antibiotics in general, at the low pH found in the stomach, compared with other antibiotics[46]. Nonetheless, patients with no history of eradication and no signs of resistance still displayed low TMRCA, implying that other events can decrease within-host diversity. These could, for example, include antibiotic treatments for any other disease, changes in gastric physiology brought about by medication, stress, or concurrent infections, and selective pressure exerted by immune responses and oxidative stress. Antibiotic-resistant populations had significantly lower TMRCAs than susceptible

populations, indicating that antibiotics have a stronger impact on population diversity than possible other uncharacterized selective events. A large reduction of genetic diversity was observed in the sequential population of patient 476, cultured from biopsies taken two years after the initial endoscopy. The patient was not treated specifically for *H. pylori* between the two endoscopies, and did not remember any treatment with antibiotics, yet the whole *H. pylori* population had become resistant to macrolides, indicating that an unrelated antibiotic regimen could have induced a population bottleneck. Through the degeneration of mutation repair systems and natural transformation, evolution has favored the accumulation of diversity in *H. pylori*, which is likely contributing to rapid adaptation to changing environmental conditions. In our case, the large amount of pre-existing genetic variation in this patient did not seem to have contributed to the rapid emergence of antibiotic resistance since the complete fixation of the adaptive 23S rRNA allele suggests that macrolides produced a hard selective sweep and a major reduction of diversity. Consequently, exposure to antibiotics, whether targeted toward eradication of *H. pylori* or not, represents an important event in the within-host evolution of these bacteria. Antibiotics are likely to have massively influenced the evolutionary dynamics of the species as a whole in recent decades as antibiotics became widely available globally and are almost ubiquitously used in almost all human populations.

The phylogenetic structure of within-host populations revealed that *H. pylori* isolates migrate between gastric regions. Migration happened frequently between the corpus and the fundus regions, thus ancestral lineages originating from the antrum were often easily identified, whereas corpus and fundus lineages were harder to distinguish from each other. *H. pylori* is a highly motile bacterium, and motility is essential for colonization;[47,48] thus its ability to move between gastric regions is not surprising. However, in contrast to its vertical motility and orientation between layers of the gastric mucus, horizontal long-distance migrations of *H. pylori* between gastric regions have not been assessed before. The data acquired in the present study hence are a leap forward in our understanding of the development of the *H. pylori* population in the whole stomach. *H. pylori* intra-stomach localization is likely to be influenced by adhesins and the availability of cell surface receptors in different regions of the stomach, and by chemotaxis, in particular, the four Tlp chemotaxis receptor proteins, which sense and react to environmental conditions such as acidity, energy levels, lactate, urea, and other stimuli[49–53]. Although corpus and fundus are anatomically distinct areas, both regions are lined by oxyntic mucosa and may thus represent similar ecological niches for *H. pylori*. Notably, parietal cells, which are responsible for acid production, are found in all gastric glands of the oxyntic mucosa but are only partially present in the antral mucosa[54]. The pH and bicarbonate gradients between the lumen and the mucus are required for spatial orientation of the bacterium and motility toward the epithelium[55,56]. A recent study demonstrated that the strength of the adherence mediated by the BabA OMP is pH-dependent, which could be a way for *H. pylori* to detect when cells have detached from the epithelial layer to be shed toward the more acidic lumen[26]. Gastric units also display distinct macroscopic organization between the two types of mucosa. Pits are deeper, glands are more branched, and the proliferative zone (isthmus) is located lower within the gland in the antrum[57]. We propose that the differences between the antral and oxyntic mucosa are limiting *H. pylori*'s ability to migrate between these regions. The variation of migration rates observed between populations is likely owing to the host environment, including disease progression. For example, patterns of gastritis are mostly dependent of the host geographical origin rather than virulence factors and lineages of *H. pylori*[58].

The isolation between the antral and oxyntic mucosa sub-populations appeared to drive independent adaptation within each niche. The independent emergence of antibiotic resistance in distinct regions observed in some patients is a clear example of such isolation and suggests that even under a global and common selective pressure, such as antibiotics, the lower frequency of migration between antral and oxyntic mucosa is hindering the selective sweep of adaptive alleles. We detected association with the antral and oxyntic mucosa in 241 genes, mainly related to chemotaxis, motility, and OMP families. With the exception of BabA and HomB, most confirmed or suspected adhesins (HopBC, HopH, HopZ, and SabA) displayed niche-specific mutations in one or more patients[59,60]. This suggests that modulating adherence to the gastric epithelium could be beneficial to improve the long-term colonization of different ecological niches. The observation that genes involved in chemotaxis and motility displayed patterns of niche-specific evolution is consistent with the important role of proper orientation and directed swimming of the bacteria for adaptation to the different types of mucosa and gastric glands. In particular, TlpB and TlpD are membrane-bound and cytoplasmic chemoreceptors, respectively, that mediate attraction to urea, repulsion from acidic pH and reactive oxygen species, and its energy taxis[49,50,52,53,61,62].

Only a minority of genes showed a signal of niche-specific adaptation in multiple patients, indicating there might be different ways to adapt to the same niche or alternatively that environmental variations between patients could be limiting the chance of convergent evolution. Moreover, different mutations were observed in different patients, the only niche-specific mutation existing identically in multiple patients was observed in the HofC protein, which is part of the second largest group of paralogous OMPs[60]. The role of this family is not well characterized; it is known that HofC is regulated by a *fur* box[63] and is able to bind lactoferrin[64]. Interestingly, HofC is required for colonization in mice[65] and was the only gene acquiring mutations during mouse adaptation in both the SS1 and SS2000 strains[66]. Lactoferrin is found in the gastric mucus of both antrum and corpus[67,68]. The gastric levels of lactoferrin are correlated with IL-8, ammonia, and inflammation levels, thus are supposed to be increased during gastritis[69]. Iron requires gastric acid for solubilization and free iron becomes less available toward the glands when bound by lactoferrin[70], thus the potentially different environments and oxyntic mucosa might drive the adaptation of HofC toward a more efficient binding of lactoferrin.

Here we showed that the within-host population structure of *H. pylori* is influenced by the physical separation of the antral and oxyntic mucosa. Consequently, it cannot be entirely dismissed that niche-specific genotypes emerged from genetic drift. Yet, functional categories relevant to host adaptation were over-represented amongst these genes and niche-specific mutations were largely non-neutral suggesting they are involved in local adaptation. Niche-specific genotypes represented only a minority of the overall diversity observed in each patient. We identified 40 high-frequency host-variable genes, which were polymorphic in > 50% of the patients. Despite belonging to similar functional categories as the niche-specific genotypes, such as the OMPs and cellular processes (regulation), these host-variable genes were not so frequently displaying polymorphisms associated to gastric regions. Unlike the niche-specific genotypes, identical mutations in host-variable genes were observed in different patients. The high in vivo mutation rate of these genes may be required for antigenic variation or adaptation to uncharacterized ecological niches. For example, *H. pylori* colonizes preferentially gastric lesions through chemotaxis[71]. Damaged gastric tissues possess different characteristics than healthy ones, and the wound-healing process is slowed down by *H. pylori*[72,73]. The diversity of

these host-variable genes means they are frequently contributing to the standing genetic variation of *H. pylori* and are targeted during selection events. For example, the plasticity of OMPs has been described many times in longitudinal isolates as well as in geographically distinct strains. The high within-host variability of some OMPs, such as HopB and HopL, suggests that variants could be selected from the donor inoculum diversity during person-to-person transmission or emerge from in situ adaptation.

## Methods

**Gastric biopsies and isolation of *H. pylori*.** Gastric biopsies were obtained from 16 patients in the Department of Gastroenterology, Hepatology and Infectious Diseases at the Otto-von-Guericke University of Magdeburg, Germany between 2012 and 2016 (Table 1). The study was performed in accordance with the Declaration of Helsinki and received the approval from the ethics committee at the Otto-von-Guericke University, Magdeburg, Germany (protocol number 80/11) as well as written informed consent from all subjects. The histological assessment was performed according to the OLGA/OLGIM staging system. *H. pylori* single colonies were isolated on blood agar plates[74] by spreading the samples directly onto the media without pre-processing.

**DNA extraction, genome sequencing, assembly, and annotation.** Genomic DNA was extracted using Genomic-tip 100/G columns (Qiagen, Hilden, Germany). All isolates were sequenced with >30-fold coverage on an Illumina MiSeq instrument using Nextera XT libraries, 50 samples dual-indexes multiplexing and 2 × 300 cycles v3 reagent cartridges. Paired-end reads were assembled de novo using SPAdes 3.9.0[75] with default parameters. In addition, selected isolates were resequenced on a PacBio *RSII* or *Sequel* instrument in order to generate completely closed reference sequences. Genomes were annotated using Prokka 1.7[76] with the *H. pylori* species database and ncRNAs were identified using Infernal[77] and Rfam 11.0[78]. Annotations of OMPs were manually curated using the 26695 and J99 reference strains as well as multiple sequence alignment and phylogenetic comparisons. For each patient, reference genomes and raw read data were submitted to NCBI under BioProject PRJNA490474. GFF Annotations files were deposited on FigShare doi:10.6084/m9.figshare.7188239.

**Antibiotic resistance testing.** Antibiotic resistance for selected isolates was tested for MTZ, CLR, and CIP using Etest strips (Liofilchem, Roseto degli Abruzzi, Italy). EUCAST clinical breakpoints values were used to determine susceptibility (The European Committee on Antimicrobial Susceptibility Testing. Breakpoint tables for interpretation of MICs and zone diameters. Version 8.1, 2018. http://www.eucast.org).

**Bioinformatics.** For each patient, a whole-genome reference-based multiple alignment was created using BWA-MEM 0.7.12[79]. Closed genomes from respective patients were used for each alignment as references. Overlapping contigs were stitched together and regions with incomplete coverage were replaced by Ns. The resulting multiple alignments were used as input for the analysis described below. Initial phylogenetic trees were generated using RAxML 8.2.9[80] with the GTR GAMMA model, 100 bootstrap replicates and the rooting algorithm option. Recombination events were predicted using ClonalFrameML 1.0[37] with default parameters, and the phylogenetic trees were corrected accordingly. Based on these recombination-corrected phylogenies, the TMRCA was calculated using a synonymous mutation rate of $1.38 \times 10^{-5}$ per site per year as determined previously[19]. For Fig. 1a, a global tree was created with all the isolates using only SNPs obtained from alignments to the reference strain 26695 and FastTree 2.1.5[81]. Figure 1b was generated with the RCircos R package[82].

Migration rates between gastric regions were computed by marginal state reconstruction using the ace function from the ape 4.1 R package[83], which implements maximum likelihood ancestral state reconstruction[84]. Ultrametric time-scaled trees were generated as described[85] and used as input; they had to be slightly adjusted to meet the ace function requirements: multichotomies were resolved as dichotomies with the multi2di function and branches with a length of 0 were change to an insignificant length of $10^{-6}$ times the total tree length. Models with equal rates (one parameter), symmetric rates (three parameters), and all rates different (six parameters) were tested consecutively by order of complexity. The more complex models were only selected if they provided a significantly higher likelihood (likelihood-ratio test, $p < 0.05$). A meristic model, in which migration between antrum and fundus is required to happen in two steps via the corpus, was also assessed using the fitDiscrete function from the geiger 2.0.6R package[86]. In order to estimate migration rates based on the within-host trees of all individuals simultaneously, a global tree was constructed by concatenating individual time-scaled trees using long (50 years) separating branches. The number of migration events was estimated by stochastic character mapping using the make.simmap function from the phytools 0.6 R package[87], which is based on a Markov chain Monte Carlo approach[88]. Character evolution was simulated for 1000 iterations using a symmetric model and 1000 simulations for each ultrametric time-scaled tree. The distribution of each type of migration event was then averaged over all trees.

Genotypes associated with gastric regions were identified using a Fisher's exact test with $p$ values corrected for multiple testing using the FDR approach[89]. To account for the high migration rate between corpus and fundus, isolates from these stomach parts were grouped together. We also accounted for recent migration events in the association analysis as follows. Using the ancestral reconstruction results, the location of each isolate was re-assigned to the location of the corresponding ancestral node with a maximum divergence time of 6 months for all patients (except patient 24 for which the maximum divergence was 1 year). Genotypes with an adjusted $p$ value < 0.01 were considered significantly associated. Enriched functional categories for genes displaying niche-associated genotypes were detected using a Chi-Square test ($p < 0.05$) and presented in Fig. 4. For these genes, departure from neutrality was tested subsequently using a McDonald–Kreitman test[35]. In brief, the four counts (Pn, Ps, Dn, Ds) of between-host (D) vs within-host (P), and non-synonymous (n) vs synonymous (s) substitutions were computed and tested for neutrality using a Fisher's exact test (FDR corrected $p$ values < 0.05). The NI was calculated as follows NI = (Pn/Ps)/(Dn/Ds).

**Reporting summary.** Further information on research design is available in the Nature Research Reporting Summary linked to this article.

## Data availability

The data that support the findings of this study are available from the article and Supplementary Information files. Moreover, reference genomes and raw sequencing reads that support the findings of this study have been deposited in the NCBI database with the accession code PRJNA490474. GFF annotation files are available in the FigShare repository with the identifier https://doi.org/10.6084/m9.figshare.7188239. The source data underlying Figs. 1a, 2a–b, and 3a–b, and Supplementary Fig. 2a are provided as a Source Data file.

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

## Acknowledgements

We thank Christine Josenhans for valuable discussions while the project was progressing as well as critical reading and commenting of the manuscript. We thank Cathrin Spröer and Boyke Bunk for performing SMRT sequencing and genome assemblies as well as Birgit Brenneke, Friederike Kops, Simone Severitt, and Nicole Heyer for their excellent technical assistance. Funding was provided from the Deutsche Forschungsgemeinschaft (grants SFB900/A1 and SFB900/Z1 to S.S.).

## Author contributions

F.A. and S.S. designed the experiments. F.A., S.W., G.P., and R.C.B performed the experiments. C.S. and P.M. carried out the clinical part of the study and provided gastric biopsies. J.O. provided PacBio sequencing. F.A., X.D., and S.S. analyzed the data. F.A., X.D., and S.S. wrote the paper. All authors provided critical comments on the manuscript and approved the final version.

## Additional information

**Competing interests:** The authors declare no competing interests.

