## [Peer Review File · Nature Communications]

Reviewers' Comments:

Reviewer #1:

Remarks to the Author:

The manuscript by Ailloud et al. describes the within-host diversity of *Helicobacter pylori* in 16 naturally infected adults and from the genomic data draws conclusions about evolutionary relationships between the isolates, how the bacteria might have migrated between the different anatomical locations in the stomach and how they have adapted to these locations. This work is expanding the knowledge about *H. pylori* biology and genome dynamics on the micro-level, something that has been reported previously but never in such an extensive dataset, nor with multiple isolates from the different gastric anatomical and physiological environments. This collected dataset has allowed the authors to not only make a descriptive report but also to more in detail test the behaviours of *H. pylori* statistically when it comes to the occurrence of between-niche migration, gene-level adaptation and population divergence and evolution within its host. One of the conclusions in this regard is that there is a quite substantial variation in the intra-host diversity and e.g. time to most recent common ancestor, which is important and should be studied further in a longitudinal manner. In summary I think this manuscript is very interesting and addresses a fundamental matter in the field, and the findings reported raises important aspects to consider for example in studying the epidemiological coupling between *H. pylori* genotype and disease.

The manuscript is based on whole-genome sequencing of ten isolates per anatomical region (fundus, corpus, and antrum), in total 440 isolates, of which the majority was sequenced on the Illumina platform resulting in incomplete draft genomes. However, a selection of isolates from each individual was also sequenced on the PacBio platform, which enabled complete genome assemblies and these genomes were subsequently used as references for alignment of the isolates within each respective individual, which is a suitable methodological approach. The methods are well described making it easy to follow how the analyses were performed.

Suggestions/revisions:

Methods:

1. The tree in Figure 1 contains all isolates. As far as I can see the methods section only describes how the within-patient trees were generated and not how the alignment for this tree was performed (or to which reference).
2. Did you make any attempt to verify/validate the synonymous mutation rate per site that was used in the calculations?

Results and discussion:

1. Figure 1A: What do the colours in the end of the branches represent? Also, those coloured dots are very hard to distinguish with the current size/resolution of the figure.
2. Figure 1B: Could the individuals represented by the rings (from outer to inner) be noted in a panel in the figure or in the legend?
3. Figure 1B: Are really a bunch of the isolates missing the 23S gene or is it a region just adjacent to the 23S gene?
4. HopB and HopC are (even more?) well-known as AlpB and AlpA. This should be clarified at some

point in the text and/or in the tables where they occur.

5. Table 2 and 3: According to the 26695 annotation curated by Resende et al. 2013 HP1117 is annotated "Cysteine-rich protein X, hcpX", HP1039 "O-antigen polymerase" and HP0567 "Autoinducer-2 Exporter (AI-2E)".

6. The section starting at line 266; "Genetic association with gastric regions..." has a semantic issue, I think. You are testing if certain genotypes are associated with the different gastric niches and you identify which polymorphisms that are associated with antrum and corpus/fundus respectively. These are located in a total of 232 genes. Here the complication begins since you from this point start referring to these genes as themselves associated with gastric regions. This is not really the case, right? Those genes have genotypes/polymorphisms that are associated with the gastric regions but they are not region-specific genes, are they? A region-specific gene would be a gene that in an accessory genome analysis was found to be associated with one niche rather than the other and as far as I understand, no such analysis was made. This way of phrasing it (that the genes are region-specific rather than that they contain region-specific polymorphisms) is also used throughout the rest of the manuscript.

7. Line 300; why do you exclude recombined polymorphisms in this analysis? These should also be under selection pressure and be able to contribute to the long-term niche adaptation. You also mention in the discussion at line 458 that "the large amount of diversity imported from a secondary infection is subsequently filtered by selection". If the reasons are technical, is there another way of analysing these sites separately?

8. Line 324: You write that the I206V mutation in HofC is associated with gastric regions in two of the individuals. Which gastric region and is it the same in the two, and if not, is it worth to highlight? Similar phrasing is used at several places in the manuscript (sometimes together with the issue noted in point 6) and I think that if the polymorphisms are indeed associated to the same niche in several individuals, that should be noted. If they are associated to corpus/fundus in some and antrum in some individuals, is it a biologically relevant finding?

9. Line 330. HcpD and HcpE are annotated as Beta-lactamases (EC-number 3.5.2.6)

10. Corpus atrophy is making the tissue and microenvironment more antrum-like ("pseudopyloric metaplasia"). I understand that the individuals in your does not give enough power to assess this statistically, but could you see any tendencies to an increased antrum-corpus migration in the individuals with higher OLGA -scores?

11. It would be interesting to look into within-host and/or within-niche accessory genome analysis using all the assemblies to see if there are any differences in genetic content between the isolates within the same individual.

12. In your reference-based mappings, did you also look into intergenic regions to see if there were any differences eg in regulatory elements between the niches?

Editorial comments:

Line 210; are should be is.

Line 320-321 "(Chi-square, $p < 0.05$)" is repeated twice

Line 324: Rephrase "part of the hyper-variable genes" as "overlapping with the hyper-variable genes"?

Line 808: "p=0.05" should maybe say "* represents p<0.05" or similar
Line 842: "striped" should say "grey"?

Reviewer #2:

Remarks to the Author:

The manuscript represents an important set of data on H pylori genetic diversity within patients. The MS is well-written and the study is generally robust, although I think their analysis needs to be tightened up to support one of their key conclusions regarding niche-specific adaptation. As it stands, I don't think they can reject the null hypothesis that the patterns they observed result from drift rather selection.

Major point:

Line 273 and elsewhere. A key point is the interpretation of niche-specific adaptation. Surely you need to show that some mutated genes are found in one gastric region but not another. And crucially, this pattern needs to be repeatable across patients. Supported by a statistical test.

Minor points:

Lines 238, 241, 276 'significantly' what are the test statistics and p-values to support this?

line 271 66% non-synonymous substitution is a weak argument. Since all nucleotide changes in positions 1 and 2 of a codon will result in a non-synonymous change, and most in the third codon a synonymous change, this is about what is expected if there are no constraints.

Line 282 and elsewhere 'hyper-variable'. This phrase can mean a specific site with a mutational process that drives genetic variation -- eg VSG proteins in Trypanosoma. I think the authors just mean there's a lot of variability in these genes. I'd pick another phrase.

Table 3 header "associated with" is more appropriate than "promoting" (the latter is an interpretation).

Reviewer #1 (Remarks to the Author):

The manuscript by Ailloud et al. describes the within-host diversity of *Helicobacter pylori* in 16 naturally infected adults and from the genomic data draws conclusions about evolutionary relationships between the isolates, how the bacteria might have migrated between the different anatomical locations in the stomach and how they have adapted to these locations. This work is expanding the knowledge about *H. pylori* biology and genome dynamics on the micro-level, something that has been reported previously but never in such an extensive dataset, nor with multiple isolates from the different gastric anatomical and physiological environments. This collected dataset has allowed the authors to not only make a descriptive report but also to more in detail test the behaviours of *H. pylori* statistically when it comes to the occurrence of between-niche migration, gene-level adaptation and population divergence and evolution within its host. One of the conclusions in this regard is that there is a quite substantial variation in the intra-host diversity and e.g. time to most recent common ancestor, which is important and should be studied further in a longitudinal manner. In summary I think this manuscript is very interesting and addresses a fundamental matter in the field, and the findings reported raises important aspects to consider for example in studying the epidemiological coupling between *H. pylori* genotype and disease.

The manuscript is based on whole-genome sequencing of ten isolates per anatomical region (fundus, corpus, and antrum), in total 440 isolates, of which the majority was sequenced on the Illumina platform resulting in incomplete draft genomes. However, a selection of isolates from each individual was also sequenced on the PacBio platform, which enabled complete genome assemblies and these genomes were subsequently used as references for

alignment of the isolates within each respective individual, which is a suitable methodological approach. The methods are well described making it easy to follow how the analyses were performed.

We thank the reviewer for the insightful and constructive review and helpful suggestions. We hope that we could fully address the questions raised in the following letter and the manuscript.

Suggestions/revisions:

Methods:

1. The tree in Figure 1 contains all isolates. As far as I can see the methods section only describes how the within-patient trees were generated and not how the alignment for this tree was performed (or to which reference).

We thank the reviewer for pointing this out. This tree was a late addition to the manuscript and the relevant details were not included in the methods. The relevant information has now been added to the methods section (l. 555-557).

2. Did you make any attempt to verify/validate the synonymous mutation rate per site that was used in the calculations?

The dataset presented in this study does not allow to determine the mutation rate, because only a single time point was sampled for 15 out of 16 individuals. The only patient for whom biopsies from a second time point were sampled, such that a mutation rate could be calculated, is patient 476. However, as described in the manuscript, the population from this individual underwent a massive bottleneck preventing us to calculate a mutation rate. The *in vivo* mutation rate of *H. pylori* has been calculated in several previous studies, using sequential isolates from chronically infected individuals, as well as strains from human volunteer challenge studies, with very similar results. The mutation rate used in our study is therefore based on robust evidence from multiple published studies (Morelli et al., PLoS Genet. 2010, Kennemann et al., PNAS 2011, Linz et al., Nat. Commun. 2014, Nell et al., Gastroenterology 2018).

Results and discussion:

1. Figure 1A: What do the colours in the end of the branches represent? Also, those coloured dots are very hard to distinguish with the current size/resolution of the figure.

We thank the reviewer for pointing out that this important information was missing in the figure legend. Colours represent the source of the individual *H. pylori* isolates, which were cultured from antrum, corpus, or fundus biopsies. However, we agree that these dots are hard to read and are also not really essential to the figure. Fig. 1A has been replaced with an updated visualization of the same tree.

2. Figure 1B: Could the individuals represented by the rings (from outer to inner) be noted in a panel in the figure or in the legend?

The legend now specifies which ring corresponds to which individual.

3. Figure 1B: Are really a bunch of the isolates missing the 23S gene or is it a region just adjacent to the 23S gene?

The description used for the black regions in this figure legend was not quite correct and we thank the reviewer for pointing this out. The term “absent” has been replaced by “not fully covered”. Regions from the reference 26695 reference strains which are not completely covered in this figure can correspond to i) missing genes such as observed in the *cagPAI* region or ii) unassembled regions due to well-known limitations of the Illumina technology (e.g. read length unable to span repetitive regions). Because the 23S gene is duplicated in *H. pylori*, both copies are sometimes not properly assembled resulting in the “uncovered” regions in this figure. A short sentence has been added to the legend of Figure 1B to clarify this.

4. HopB and HopC are (even more?) well-known as AlpB and AlpA. This should be clarified at some point in the text and/or in the tables where they occur.

We thank the reviewer for this suggestion and have added the names AlpA and AlpB in the text where HopB and HopC are first mentioned (l. 214) as well as in tables 2 and 3.

5. Table 2 and 3: According to the 26695 annotation curated by Resende et al. 2013 HP1117 is annotated “Cysteine-rich protein X, *hcpX*”, HP1039 “O-antigen polymerase” and HP0567 “Autoinducer-2 Exporter (AI-2E)”.

The annotations have been updated appropriately in the text (l. 273), Table 2, 3 and Supplemental Table 3.

6. The section starting at line 266; “Genetic association with gastric regions...” has a semantic issue, I think. You are testing if certain genotypes are associated with the different gastric niches and you identify which polymorphisms that are associated with antrum and corpus/fundus respectively. These are located in a total of 232 genes. Here the complication begins since you from this point start referring to these genes as themselves associated with gastric regions. This is not really the case, right? Those genes have genotypes/polymorphisms that are associated with the gastric regions but they are not region-specific genes, are they? A region-specific gene would be a gene that in an accessory genome analysis was found to be associated with one niche rather than the other and as far as I understand, no such analysis was made. This way of phrasing it (that the genes are region-specific rather than that they contain region-specific polymorphisms) is also used throughout the rest of the manuscript.

What is described as “Genetic association with gastric regions” indeed refers to polymorphisms. We clarified this point throughout the manuscript. The question of “actual” region-specific gene is discussed in Comment #11 later in this response.

7. Line 300; why do you exclude recombined polymorphisms in this analysis? These should also be under selection pressure and be able to contribute to the long-term niche adaptation. You also mention in the discussion at line 458 that “the large amount of diversity imported from a secondary infection is subsequently filtered by selection”. If the reasons are technical, is there another way of analysing these sites separately?

We thank the reviewer for bringing this up. The patterns of imports observed within-host (described in the text and Supplemental Figure S3) indeed strongly suggest that recombined polymorphisms are under selection pressure.

However, only 20% of the individuals included in this analysis showed any evidence of recombination with an unrelated strain during mixed infection. Moreover, because of the high genetic diversity of *H. pylori*, a single import from another strain is likely to transfer multiple polymorphisms. Because it is unlikely that all recombined polymorphisms from a

single import contribute to adaptation, including those in the analysis might artificially inflate the number of niche-associated polymorphisms in populations containing imports, and increase noise. In order to draw robust conclusions from the dataset, we therefore excluded imported polymorphisms.

8. Line 324: You write that the I206V mutation in HofC is associated with gastric regions in two of the individuals. Which gastric region and is it the same in the two, and if not, is it worth to highlight? Similar phrasing is used at several places in the manuscript (sometimes together with the issue noted in point 6) and I think that if the polymorphisms are indeed associated to the same niche in several individuals, that should be noted. If they are associated to corpus/fundus in some and antrum in some individuals, is it a biologically relevant finding?

As mentioned in our response to comment #6, genetic association with gastric regions always refers to antrum versus corpus+fundus. In the case of HofC, we observed in two individuals (patients 13 and 20) that the antrum isolates had a valine at position 206 while the corpus isolates had an isoleucine at this position. Whether this conservative amino acid exchange is biologically relevant, is unknown. This sentence has been clarified (lines 264-266).

9. Line 330. HcpD and HcpE are annotated as Beta-lactamases (EC-number 3.5.2.6)

HcpD and HcpE do have sequence similarities with HcpA, which has been described as having some penicillin-binding and beta-lactamase activity (Mittl et al. 2000, Luthy et al. 2002). Nonetheless, to our knowledge, beta-lactamase activities have not been properly functionally characterized for HcpD and HcpE, nor have these genes been associated with beta-lactam resistance. Consequently, we decided to use more careful annotations for these genes/proteins.

10. Corpus atrophy is making the tissue and microenvironment more antrum-like ("pseudopyloric metaplasia"). I understand that the individuals in your does not give enough power to assess this statistically, but could you see any tendencies to an increased antrum-corporum migration in the individuals with higher OLGA -scores?

We looked into this hypothesis, but did not observe such a tendency in this dataset. Even with a larger set of naturally infected individuals, we believe that it would be hard to detect any correlation between migration and disease progression considering how frequently population bottlenecks seems to happen.

11. It would be interesting to look into within-host and/or within-niche accessory genome analysis using all the assemblies to see if there are any differences in genetic content between the isolates within the same individual.

We thank the reviewer for this comment. We did analyze the dataset for differential gene content within-host. Overall, gene content was very similar between isolates from one individual, with few exceptions, none of which showed any association with stomach niches. The most significant difference of gene content was the differential presence/absence of the cagPAI in patient 476, which is described in the first results paragraph. However, it was not possible to link the cagPAI status to a specific niche in this patient due to i) the extensive mixing of isolates and ii) the absence of fundus isolates (see the "1st endoscopy" tree in Fig. 4).

12. In your reference-based mappings, did you also look into intergenic regions to see if there were any differences eg in regulatory elements between the niches?

We thank the reviewer for this comment, and agree that selection might act on intergenic regions in a similar way than on coding sequences. We did not look specifically at polymorphisms located in intergenic regions and rather chose to focus on non-synonymous polymorphisms resulting in a change in a protein sequence. The main reason was that even if a regulatory sequence is known, such as a promoter element or small RNA, it is in most cases impossible to predict the phenotype resulting from any single basepair change.

Editorial comments:

Line 210; are should be is.

Corrected (l. 145).

Line 320-321 "(Chi-square, $p < 0.05$)" is repeated twice

Corrected (l. 257).

Line 324: Rephrase "part of the hyper-variable genes" as "overlapping with the hyper-variable genes"?

Rephrased accordingly as it makes the sentence clearer (l. 267).

Line 808: " $p = 0.05$ " should maybe say "*" represents $p < 0.05$ " or similar

Changed accordingly (l. 928).

Line 842: "striped" should say "grey"?

Indeed. Changed accordingly (l. 962).

Reviewer #2 (Remarks to the Author):

The manuscript represents an important set of data on *H pylori* genetic diversity within patients. The MS is well-written and the study is generally robust, although I think their analysis needs to be tightened up to support one of their key conclusions regarding niche-specific adaptation. As it stands, I don't think they can reject the null hypothesis that the patterns they observed result from drift rather selection.

We thank the reviewer for the insightful and constructive review and helpful suggestions. We hope that we could fully address the questions raised as specified below, and in the manuscript.

Major point:

Line 273 and elsewhere. A key point is the interpretation of niche-specific adaptation. Surely you need to show that some mutated genes are found in one gastric region but not another. And crucially, this pattern needs to be repeatable across patients. Supported by a statistical test.

L. 273 (now l. 207) and the first paragraph of this section are actually not yet about niche-specific adaptation to the antrum, corpus and fundus. This section rather serves as a more general starting analysis to determine whether we can identify genes that mutate more frequently within-host. In this analysis, we showed that a set of 40 genes appeared to

mutate more frequently than expected by performing a random sampling simulation using the observed number of variants within each population. In order to improve the structure of the results section, we have now separated this paragraph from the following text and provided it with a separate heading.

The second paragraph is concerned with niche-specific adaptation. As mentioned by the reviewer, the observed patterns of niche-specific polymorphisms could also be attributed to drift due to the underlying structure of *H. pylori* populations within the human stomach. Modern association studies usually attempt to control for such effects. However, in the context of this study, the characters (i.e. the stomach regions) we looked to associate with specific polymorphisms represent distinct geographical environments and thus are expectedly responsible for stratification of the within-host populations.

One way to compensate for such drawback would have been, indeed, to prove that the niche-specific mutations we observed emerged under positive selection. However, inferring natural selection using classical tests was also challenging for several reasons. The size of our dataset made it unlikely to observe such niche-specific polymorphisms in many patients simultaneously. The number of populations we worked on was mainly limited because it required isolating *H. pylori* again from "fresh" biopsies, because existing strain collections do not contain multiple *H. pylori* clones per biopsy and thus do not permit an assessment of within-host population structure. Despite the fair amount of diversity observed in some populations, the number of polymorphisms observed at the scale of single gene is still limited. In many of our candidates, the niche-specific mutation is actually the only polymorphism observed in the gene for a given population.

Nonetheless, we identified significantly enriched functional categories within the genes containing niche-specific polymorphisms, indicating that specific groups of genes are more likely to display niche-specific mutations. Moreover, these functional categories included many outer membrane proteins which have been shown repeatedly as major contributors to host-interaction as well as chemotaxis, motility and regulation related proteins which would be likely relevant in a scenario where a bacterium has to adapt to compartments with variable physiological conditions such as the gastric regions. In order to provide additional evidence that the mutations observed in these genes are strong candidates for niche-adaptation we performed a McDonald-Kreitman test on all the genes listed in Table 3. The McDonald-Kreitman test is usually performed to compare inter- versus intra-species diversity, but here we compared inter- versus intra-host diversity. 31 out of 47 genes had a FDR-adjusted p-value < 0.05 (Fisher's exact test), the neutrality indexes could be calculated for 26 out these 31 genes and were > 1. Overall these results show that the within-host polymorphisms observed in these genes are non-neutral. Here, neutrality indexes > 1 indicate an excess of within-host polymorphism. In the context of this study, these mutations are fixed within local sub-populations, thus the excess of within-host populations can be interpreted as a sign of local adaptation. However, in the regular context of the McDonald-Kreitman test, these mutations could be interpreted as slightly deleterious mutations (which would be purged during transition to new hosts on a global evolutionary scale). Again, both interpretations cannot be fully disentangled. These new results were added to the manuscript and in Table 3.

Minor points:

Lines 238, 241, 276 'significantly' what are the test statistics and p-values to support this?

Line 238: significance is based on 95% confidence intervals of the maximum likelihood estimates of the discrete character reconstruction. Details have been added to the text (l. 173)

Line 276: significance is based on 95% confidence intervals of 10,000 replications from a random sampling simulation (see Supplemental Figure 2). Details have been added to the text (l. 211)

Line 241: the meristic and symmetric model were compared using the Akaike information criterion, AICc values have been added for both models (l. 177).

line 271 66% non-synonymous substitution is a weak argument. Since all nucleotide changes in positions 1 and 2 of a codon will result in a non-synonymous change, and most in the third codon a synonymous change, this is about what is expected if there are no constraints.

We agree with the reviewer's comment. The sentence was deleted.

Line 282 and elsewhere 'hyper-variable'. This phrase can mean a specific site with a mutational process that drives genetic variation -- eg VSG proteins in Trypanosoma. I think the authors just mean there's a lot of variability in these genes. I'd pick another phrase.

We agree that this phrase could lead to confusion. It has been changed to "host-variable genes" with an additional definition as "genes displaying a high-frequency of within-host variation".

Table 3 header "associated with" is more appropriate than "promoting" (the latter is an interpretation).

We agree and have modified the header accordingly.

Reviewers' Comments:

Reviewer #1:

Remarks to the Author:

I have carefully read the revised manuscript and I think the authors have adequately addressed each point of the reviewers comments and that this interesting manuscript may now be accepted for submission.

Very minor point; is the comma at line 43 in the abstract really intended? It gives the sentence a weird flow.

Reviewer #2:

Remarks to the Author:

Overall, the authors have dealt with my main comment of separating selection from drift using the MacDonal-Kreitman test. Nevertheless the text does not accurately reflect where there are genetic differences associated with location versus where this has been shown to be due to selection. Especially in the results, I would like to see a clearer separation of what the data shows with respect to patterns of diversity, versus what it shows with respect to the processes that generate those patterns. By the authors' admission in the rebuttal letter, the only evidence for adaptation comes from the MK tests.

(1) line 157. This section only describes migration and not 'niche-specific adaptation'. Similarly line 43 in the abstract.

In the rebuttal they say: 'However, in the context of this study, the characters (i.e. the stomach regions) we looked to associate with specific polymorphisms represent distinct geographical environments and thus are expectedly responsible for stratification of the within-host populations.' This is a reasonable hypothesis, but genetic differences in different locations does not itself demonstrate adaptation.

(2) line 211. Showing high levels of non-synonymous SNPs at some genes could reflect adaptation, higher mutation rate or lack of functional constraints at those genes. The randomisation shows that some parts of the genome acquire more substitutions than others, not the cause of these differences, ie this does not show adaptation.

Reviewer #1 (Remarks to the Author):

I have carefully read the revised manuscript and I think the authors have adequately addressed each point of the reviewers comments and that this interesting manuscript may now be accepted for submission.

Very minor point; is the comma at line 43 in the abstract really intended? It gives the sentence a weird flow.

This sentence has been removed in order to meet the length requirements of the abstract.

Reviewer #2 (Remarks to the Author):

Overall, the authors have dealt with my main comment of separating selection from drift using the MacDonald-Kreitman test. Nevertheless the text does not accurately reflect where there are genetic differences associated with location versus where this has been shown to be due to selection. Especially in the results, I would like to see a clearer separation of what the data shows with respect to patterns of diversity, versus what it shows with respect to the processes that generate those patterns. By the authors' admission in the rebuttal letter, the only evidence for adaptation comes from the MK tests.

The section about niche-specific mutations has been reorganized and slightly rewritten to better explain which genes actually show signs of adaptation according to the results of the MK test.

(1) line 157. This section only describes migration and not 'niche-specific adaptation'. Similarly line 43 in the abstract.

In the rebuttal they say: 'However, in the context of this study, the characters (i.e. the stomach regions) we looked to associate with specific polymorphisms represent distinct geographical environments and thus are expectedly responsible for stratification of the within-host populations.' This is a reasonable hypothesis, but genetic differences in different locations does not itself demonstrate adaptation.

The title of the section L157 has been changed to better reflect its content. The abstract has been rewritten as well. We agree that the mere presence of genetic differences in different locations does not demonstrate adaptation per se. This hypothesis is then supported in the text by the functional enrichment analysis, which suggests that these differences have a tendency to accumulate in sets of genes with similar functions, and furthermore by the McDonald-Kreitman test added during the revision process which provide evidence of positive selection.

(2) line 211. Showing high levels of non-synonymous SNPs at some genes could reflect adaptation, higher mutation rate or lack of functional constraints at those genes. The randomisation shows that

some parts of the genome acquire more substitutions than others, not the cause of these differences, ie this does not show adaptation

The sentence L211 should have been removed as stated in our previous rebuttal letter since we agreed with the reviewer's previous comment that the observation about the overall high level of non-synonymous SNPs in populations does not indicate adaptation. We apologize for the mistake. It has now been properly removed. Moreover, adaptation is only mentioned in this paragraph to describe the possibility of parallel adaptation in the case where identical mutations were frequently observed in multiple patients, which would indicate the existence of some functional constraint.